# Ethical, Legal, Organisational and Social Issues of Teleneurology: A Scoping Review

**DOI:** 10.3390/ijerph20043694

**Published:** 2023-02-19

**Authors:** Alezandra Torres-Castaño, Analía Abt-Sacks, Ana Toledo-Chávarri, José Carlos Suarez-Herrera, Janet Delgado-Rodríguez, Beatriz León-Salas, Yadira González-Hernández, Montserrat Carmona-Rodríguez, Pedro Serrano-Aguilar

**Affiliations:** 1Canary Islands Health Research Institute Foundation (FIISC), 38320 Tenerife, Spain; 2Evaluation Unit of the Canary Islands Health Service (SESCS), 38109 Tenerife, Spain; 3Network for Research on Chronicity, Primary Care, and Health Promotion (RICAPPS), 28029 Madrid, Spain; 4The Spanish Network of Agencies for Health Technology Assessment and Services of the National Health System (RedETS), 28071 Madrid, Spain; 5UNITWIN/UNESCO Chair, Research, Planning and Development of Local Health Systems, Department of Clinical Sciences, University of Las Palmas de Gran Canaria, 35001 Las Palmas de Gran Canaria, Spain; 6Department of Philosophy I, University of Granada, 18071 Granada, Spain; 7Health Technology Assessment Agency, Instituto de Salud Carlos III, 28029 Madrid, Spain; 8Institute of Biomedical Technologies, University of La Laguna, 38200 Tenerife, Spain

**Keywords:** teleneurology, health technology assessment, ethical, legal, organizational, social, equity, accessibility

## Abstract

Background: Neurological disorders are the leading cause of disability and the second leading cause of death worldwide. Teleneurology (TN) allows neurology to be applied when the doctor and patient are not present in the same place, and sometimes not at the same time. In February 2021, the Spanish Ministry of Health requested a health technology assessment report on the implementation of TN as a complement to face-to-face neurological care. Methods: A scoping review was conducted to answer the question on the ethical, legal, social, organisational, patient (ELSI) and environmental impact of TN. The assessment of these aspects was carried out by adapting the EUnetHTA Core Model 3.0 framework, the criteria established by the Spanish Network of Health Technology Assessment Agencies and the analysis criteria of the European Validate (VALues In Doing Assessments of healthcare TEchnologies) project. Key stakeholders were invited to discuss their concerns about TN in an online meeting. Subsequently, the following electronic databases were consulted from 2016 to 10 June 2021: MEDLINE and EMBASE. Results: 79 studies met the inclusion criteria. This scoping review includes 37 studies related to acceptability and equity, 15 studies developed during COVID and 1 study on environmental aspects. Overall, the reported results reaffirm the necessary complementarity of TN with the usual face-to-face care. Conclusions: This need for complementarity relates to factors such as acceptability, feasibility, risk of dehumanisation and aspects related to privacy and the confidentiality of sensitive data.

## 1. Introduction

Neurological diseases include all disorders affecting the nervous system in general, or some of its components, such as the brain, cerebellum, brainstem, spinal cord or nerves. Overall, these disorders are the leading cause of disability and the second leading cause of death worldwide. In the past 30 years, the absolute numbers of deaths and people with disabilities owing to neurological diseases have risen substantially, particularly in low-income and middle-income countries, and further increases are expected globally as a result of population growth and ageing [1].

The increase in the population demand for care, together with territorial inequalities in patient access to health services, and the difficulties of economic sustainability for health systems in most countries, has led to telemedicine being an alternative to these three divergent challenges [2]. Unfortunately, for almost two decades, limitations in available scientific evidence delayed the widespread adoption of the different modalities of telemedicine in its different applications. Time has elapsed, accompanied by improvements in the designs of evaluative studies on telemedicine, and, more recently, the need for tools to meet the population’s health needs during the COVID-19 pandemic, which favour the implementation and diffusion of telemedicine in all health fields [3].

The practice of telemedicine (TM) has changed the way healthcare is delivered, towards a model that can be considered a “disruptive innovation” [4]. Teleneurology (TN) is a procedure that allows neurology to be performed when the doctor and patient are not present in the same place, and sometimes not at the same time [5].

The successful long-term use and outcomes of technology-based interventions (TBI) can be improved by identifying the factors that influence their adoption and developing strategies to address specific barriers and build on identified facilitators [6]. The remote provision of health services, such as TN, in particular, presents challenges at different levels, such as the political, organisational, technical, ethical-legal and clinical [4,7,8].

The use of TM in the case of neurological diseases must be sensitive to characteristics that determine the conditions of healthcare, such as the urgency of care for some of these diseases. While, once diagnosed, most neurological patients can be cared for on an outpatient basis, those requiring immediate diagnostic tests and treatment must be managed in a hospital setting [9]. The greatest evidence for the use of TN has been in neurocritical care and emergency stroke, where rapid assessment by a neurologist or other cerebrovascular pathology specialist is required, especially in remote underserved communities, such as rural areas, where direct access to neurology services are lacking [10,11,12,13,14].

While the adoption of healthcare interventions is mostly guided by evidence on its effectiveness, safety and cost-effectiveness, the impact of the ethical, legal and social issues (also known as ELSI) [15] and organizational, environmental aspects and patients’ perspectives that may derive from the use of TN is scarcely known, despite their relevance in successful implementation. The evaluation of all these dimensions should consider the type of neurological disease, the care trajectory (prevention, clinical management, clinical home follow-up, etc.) and its complementary use to usual face-to-face care. To review the evidence on effectiveness, cost-effectiveness and impacts on the ethical, legal, organisational and social dimensions of the complementary use of teleneurology, the Spanish Ministry of Health in February 2021 commissioned The Spanish Network of Agencies for Health Technology Assessment (RedETS) with the elaboration of a health technology assessment report. The objective of this scoping review was to identify the impact of the ELSI dimensions and other dimensions, such as the organisational and environmental, to analyse in depth the challenges of the implementation of teleneurology as a complement to face-to-face neurology care. This scoping review is part of the health technology assessment report mentioned before. 

To meet this objective, a methods section is included to explain the approach we followed, based on a consultation carried out in a virtual meeting with experts in the field of teleneurology. This meeting allowed us to identify key issues in the use of teleneurology, which was complemented by the development of a systematic search. Subsequently, the main themes that emerged according to the Core Model 3.0 of EUnetHTA are presented. In the discussion, the implications of these results are analysed, as well as the main strengths and limitations, and the implications for future practice. Finally, the most relevant ideas that emerged from the development of this scoping review are presented in the Conclusions section.

## 2. Methods

A scoping review (SR) was performed according to the recommended five-step framework for scoping reviews [16,17,18,19]. A detailed protocol for the review was developed by the research team and further approved by the Spanish Ministry of Health (available from the corresponding author on request). The assessment of ethical, legal, organisational and social aspects was carried out by adapting the framework of the Core Model 3.0 of EUnetHTA, the criteria established by the Spanish Network of Agencies for Health Technology Assessment for the National Health Service, and the analysis criteria of the European VALIDATE Project (VALues In Doing Assessments of healthcare TEchnologies) [20,21,22]. None of these frameworks addresses patient safety specifically in teleneurology, but rather in general health technology terms. The environmental aspects were assessed considering the analysis of the carbon footprint caused by the technology during its complete life cycle. The whole life cycle (WLC) includes five stages: (1) material acquisition and pre-processing; (2) production; (3) distribution and storage; (4) use and (5) end of [7,8].

### 2.1. Information Sources and Search Strategy

Within the VALIDATE approach, the initial key question was whether the Health Technology Assessment (HTA) addresses all relevant concerns that different stakeholders may have concerning teleneurology as a complement to face-to-face neurology care. Thus, to ensure that the perspective of all the stakeholders was included, we invited them to discuss their concerns about teleneurology in an online meeting to identify the main problems and benefits of TN from their perspectives. This group of stakeholders belonged to the working group and were collaborators in the HTA report described above. They were chosen for their experience and relevant publications in the field of TN. For the online meeting, they did not sign a specific consent form, as they had previously completed these documents as authors of the report.

This meeting was held on 6 June 2021. The participants were representatives of patient associations, representatives of scientific societies, healthcare professionals and industry representatives with expertise in neurological diseases and/or TM. The list of participants can be found in Appendix A. The consultation was guided by the following authors ATC; ATCh; JDR; BLS and PSP. A semi-structured script available in Appendix A was used for this purpose.

Patient representatives were the first to respond to each of the questions, healthcare professionals and representatives of scientific societies gave their perspective next, while industry representatives were the last ones. This pre-established order of interventions was set up to prevent patient interventions from being constrained by differences in knowledge and ease of communication on the part of professional and industry representatives. A summary of the topics discussed was sent after the meeting and reviewed by all participants. The themes raised at this meeting were translated into research questions which are shown in column 1 of Table 1. Those themes that could not be addressed in this review because insufficient evidence was found were taken as research needs. These themes are shown in column 2 of Table 1.

Subsequently, a systematic search of articles was conducted in the electronic databases MEDLINE (OVID interface) and EMBASE (Elsevier interface) from 2016 to June 10, 2021. The search strategy was initially developed in MEDLINE, using a combination of controlled vocabulary and free text terms and was then adapted for EMBASE. The full search strategy is available in Appendix A. 

Search terms included the following: teleneurology, mhealth, teleconsultation, TM complementary service, ethics, organisational, legal, social, environmental impacts and carbon footprint. Searches were limited to the English and Spanish languages. 

### 2.2. Selection Criteria

Studies were eligible for inclusion if they fulfilled the following criteria: Design: Qualitative, mixed studies, observational studies, systematic and narrative reviews covering ethical, legal, organisational, social and environmental aspects.Language: Articles published in English and Spanish. Population: People with any neurological disease (adults and children).Intervention: Teleneurology as a complementary service to face-to-face neurological care.Comparison: Regular face-to-face neurology care.Outcomes: The outcomes selected were all those that answered the research questions that emerged from the stakeholder group meeting.

### 2.3. Study Selection 

Retrieved references were first screened independently and in duplicate based on their titles and abstracts by four reviewers. The full text of the potentially eligible references was then screened again, independently and in duplicate, to confirm eligibility. Doubts and disagreements between the reviewers were resolved by discussion in pairs, and where no consensus was reached, a third reviewer was consulted.

### 2.4. Data Collection Process 

Data extraction from included studies was carried out by three reviewers. The data collection included the identification of the article (authors, publication date, country, etc.), methodology (design, intervention characteristics, participant’s characteristics, etc.) and study results. 

### 2.5. Quality Assessment 

To ascertain the quality of the studies, the methodological limitations section within each of the included studies was consulted. No specific scale was used to assess the quality of the studies.

### 2.6. Synthesis of Results

The data analysis was performed using a thematic analysis, identifying the most relevant categories that were extracted from the included studies followed by a narrative synthesis. Finally, the findings were checked against the EUnetHTA Core Model 3.0. Evaluative framework to validate their relevance.

## 3. Results

From a total of 2230 initially identified references (1605 after eliminating the duplicates), 212 were selected after the first screening round by title and abstract. After the second screening round, based on full-text revision, 79 studies met the criteria. Thirty-seven studies were related to acceptability and equity, 15 studies related to the reorganisation of neurology services as a consequence of the COVID pandemic, one study addressed environmental aspects related to technology and 26 studies were related to feasibility. Figure 1 shows the PRISMA flowchart of the study-selection process.

This scoping review will only include 53 studies: the 37 studies related to acceptability and equity, the 15 studies developed during COVID, and the study on environmental aspects.

The research question addressed was: What ethical, legal, organisational and social aspects related to the use of TN as a complementary service to face-to-face neurological care are relevant for its implementation in the health system?

### 3.1. Characteristics of Included Studies

Studies from the following countries were included: the United Kingdom, Netherlands, Norway, Australia, Sweden, Mexico, Canada, the United States of America, Uganda, Belgium, Spain and Germany. The included study designs were: 19 qualitative studies [6,23,24,25,26,27,28,29,30,31,32,33,34,35,36,37,38,39,40,41,42], 9 mixed-methods studies [24,43,44,45,46,47,48,49,50], 6 implementation reports [23,51,52,53,54,55] 6 descriptive studies [56,57,58,59,60,61], 1 cohort [62], 1 cross-sectional [63], 1 prospective cohort [64], 1 retrospective cohort [65] and 1 observational cohort (case/control) [66].

The population of these studies included patients with different neurological disorders and health care professionals: (a) stroke 20 studies [6,23,24,26,27,28,30,31,32,35,39,40,42,47,56,58,62,67,68,69], at different disease stages (acute o rehabilitation) and technological modalities (synchronous and asynchronous consultations), (b) Parkinson’s disease, 9 studies (1 about a body sensor to assess motor symptoms, 1 about smartphone use for follow-up, 1 about virtual visit, 1 about eHealth functionalities for co-care, and 1 about implementation issues) [33,34,41,45,54,57,66,70,71], (c) epilepsy, 6 studies (1 related to digital and wearable technology, 1 about mHealth and 1 through video conferencing for children in rural areas) [36,44,50,60,61,72], (d) multiple sclerosis, 2 studies (1 related to m-health in people with relapsing remitting sclerosis and progressive multiple sclerosis, and 1 on telecommunication technologies and rehabilitation services [37,46,63]; 1 dementia [64] and 2 other studies with other pathologies (patients with multiple sclerosis, depression and epilepsy, remote sensing technology for symptom severity) [38,43]. The other 11 studies analysed neurological disorders in general (1 remote follow-up, 1 adoption of virtual reality in patients with motor impairment for chronic conditions, 1 examine implementation issues, 1 virtual rehabilitation and 1 acceptability and long-term adherence) [25,29,32,48,51,53,55,59,65,73,74].

Finally, one study focused on greenhouse gas reduction using neuro-emergency telemedicine consultations [52]. Besides patients, caregivers and professionals in primary and specialized care, as well as resident physicians, were also part of the selected study population.

The main outcomes reported were about barriers and facilitators of TN use, perceptions and expectations about the use of TN, perceived benefits of its use, competence development needs, resistance from professionals, legal and organisational aspects, and ethical and environmental dimensions. Appendix A shows the characteristics of the included studies.

### 3.2. Quality Assessment 

The main methodological limitations observed in the included studies were the small sample size and the inadequate examination of the social determinants of health (gender, age, ethnicity, co-morbidity, disability, digital literacy, family/social support, geographic residence and socioeconomic status) [23,25,26,27,28,34,43,45,46,47,61,62,69]. Moreover, insufficient information is provided to assess methodological limitations [6,51,63]. Other studies were carried out in hypothetical settings [36,37,56]. Furthermore, the studies analysing TN in various health conditions do not allow us to delve into specific aspects of care for each condition, limiting the transferability of their results [25,32,38,51,58,73,74].

### 3.3. Synthesis of Results

A narrative synthesis of the main results was conducted and organised according to the research questions identified by the stakeholders. The research questions were addressed according to the EUnetHTA Core Model 3.0 framework, the criteria established by the Spanish Network of Health Technology Assessment Agencies for the National Health Service, and the themes that emerged from the included studies. Main findings were finally formulated under the following issues: (1) Acceptability, (2) Usability, (3) Ethical aspects and risk of dehumanisation, (4) Equity (universal accessibility to the technology), (5) Organisational aspects related to implementation, quality and environmental implications.

#### 3.3.1. Acceptability

Acceptability was evaluated from the point of view of patients, caregivers, community leaders and health professionals involved at different health-care levels: primary and specialized care, and emergencies [27,42,58]; including different modalities of TN, either by videoconference or through remote measurement technologies [43], such as tracking devices or mobile applications for stroke, multiple sclerosis, Parkinson’s disease, epilepsy and depression, and for different moments in the care trajectory. Patients and caregivers considered TN as an acceptable complementary option to face-to-face neurology consultation [6,34,45]. The benefits resulting from the use of TN, such as cost savings in travel or waiting time in health services and the avoidance of work absenteeism and reducing caregiver burden and fatigue, are aspects that can improve its acceptability. Ease of access to otherwise inaccessible specialists is another reason for patient acceptability [56]. Monitoring through body sensors was acceptable to patients in the short term, although, in the long term, they reported discomfort in aspects related to usability and doubts about the quality of the data captured [45].

However, studies with healthcare professionals and patients with Parkinson’s disease and stroke indicate that the quality of personal interactions is a key aspect that may condition the acceptability of TN [27,30,34]. Patients also expressed concern that TN should ensure interpersonal engagement, understood as attentive or active listening, as well as the ability to establish a bond of trust [30,34]. In the rehabilitation phase, patients find complementary TN consultations acceptable when there is prior training, when some social interaction is provided, when caregivers are available, and when consultations take place in the home setting [26]. 

From the professional perspective, the use of tele neurorehabilitation technologies is complex, considering the context-specific benefits and challenges. Some facilitator factors include the possibility of providing accurate and objective feedback and monitoring performance, as well as its potential to encourage patients to continue physical activity outside therapy hours [25]. Practitioners noted, as a requirement for acceptability, that mobile applications be previously validated by experts [46].

#### 3.3.2. Usability

Healthcare professionals stated that the different TN interventions could be combined in a central dashboard in which all facilities are directly available through a digital environment, allowing their synchronisation with the programs used for the patients’ electronic medical records [47]. In terms of user experience, health professionals prefer pictograms, symbols and graphics, and a smooth, flicker-free interface. The design configuration should be adjustable and ensure that it can be heard well, i.e., with the ability to hear the written text and to emit sounds as alerts or feedback. Simplicity, as measured by the limited number of web pages that are opened as a result of using a service, as well as by the information provided on a single screen, should be ensured [38]. In addition, the TN tool for rehabilitation should have a support service, a menu with frequently asked questions, videos with instructions for use, a help desk or direct assistance at home or the workplace, and the monitoring of physical activity and health status, as well as the possibility of having personalised goals, socialisation and gamification options [38,46,56,63]. To enhance the video consultation experience for both patients and professionals, proposed solutions included virtual waiting rooms and multi-user interfaces to facilitate multidisciplinary care [47].

#### 3.3.3. Ethical Aspects and Risk of Dehumanisation

As in the face-to-face modality of care, all ethical standards and integrity of medical practice must be maintained in TN to ensure that patients receive the best possible healthcare. All remote consultations should be treated with a confidential guarantee [72], including the storage of data using mobile and wearable technology to monitor health outcomes, subject to signed consent [71]. Patients and caregivers must receive adequate information to ensure that the identity and credentials of remote professionals are explicit to them [30,67,72].

Healthcare professionals have stated that TN should provide the standards they need to be comfortable. While space, adequate technology implementation and organisational support are all necessary for the successful use of TN, they are not enough for its ethical delivery.

Other studies warn about ethical, legal and technological implications of the use of mobile send-tracking apps for persons affected with Parkinson’s disease or stroke [67,71] because is necessary to maintain an adequate balance between the proper monitoring of patients for strictly therapeutic purposes and excessive surveillance of the person, which may violate their privacy. It is important to provide enough information to allow patients to choose whether they want to measure aspects of their health and well-being and avoid collecting data without an explicit purpose, to reduce the emotional burden that it can mean for them [71]. Careful co-design and feedback with users must be respected throughout the process, to ensure that the technology is adapted to the user’s needs and preferences, as well as reporting all interactions with non-human tools and resources (e.g., chatbot) and even ensuring the signing of consent-to-participate forms [71].

The paradigm shift in routine medical care, related to the ability of professionals to convey compassion, should also be incorporated into TM interventions, especially because the forms of expression of empathy in virtual meetings differ from those in person. This communication dimension is reported in studies about paediatric neuropsychology services during the COVID-19 pandemic [49] and clinical encounters for acute stroke. The feeling of deterioration or rupture of the traditional health professional–patient relationship constitutes a major concern for patients and companions/caregivers and the health professionals themselves, due to the risk of dehumanisation that ICT-mediated therapeutic relationships entail, both in ictus healthcare [30,58,62,72] as well as autism, with paediatric and adult populations [30,46,58,72,75]. In Parkinson’s disease, the quality of interpersonal engagement is identified as one major theme for patients’ perceptions of virtual visits [34]. Among health professionals, there is a risk of dehumanisation related to examining patients without physical contact [34]. 

#### 3.3.4. Equity

TN promises to increase access to specialised care, contributing to reducing healthcare inequities associated with accessibility in remote or rural areas, and patient movement difficulties. Indeed, this avoid a loss of productivity for the accompanying caregiver too [49,55,55]. Besides the potential to improve equity in access to evaluation, diagnosis, and follow-up, TN reduces barriers such as travel costs and community stigma [72]. Moreover, the complementary use of videoconferencing could have relatively fewer restrictions than in-person visits, which are often scheduled months in advance, and could therefore reduce wait times.

Several studies show also the interest of patients in telerehabilitation, because, in addition to reducing the burden of access to health care, it reduces periodic transportation, besides being more comfortable at home [26,34,42,72]. However, obstacles to universal accessibility to TN have been identified, the main ones being the digital divide, and the diverse functional capacity of users, especially the elderly or those with some disabilities such as multiple sclerosis [63] that are not contemplated in the design and implementation of these technologies. People with different moderate and severe physical or cognitive disabilities who, in principle, could benefit more from telerehabilitation programs, have, on average, more difficulties in accessing and using technologies [63]. For patients with chronic progressive neurological conditions and loss of muscle function for whom face-to-face consultations become increasingly challenging, video visits have been found to improve the patient experience, although several challenges remain [59]. The most vulnerable people with the greatest needs are often the most difficult to serve virtually, not only because of their diagnosis (patients with sensory or cognitive deficits) but also because of the need to have the support of a companion/caregiver and their socioeconomic status and consequent difficulties in access to internet services [27,48,57]. The main barriers to accessing the telematic sessions were socioeconomic and linguistic disadvantages—difficulties in communicating in the language of the professionals or poor language skills—together with family health problems [49]. Due to these barriers, health professionals consider that the complementary use of TN should only be offered to people who are willing to be treated under this modality and have the required skills [58].

In European countries such as Belgium or France, where TM reimbursement has been regulated, teleconsultations are voluntary and require patient consent [76]. To ensure effective accessibility and an adequate quality of TN, the necessary technical support to patients should be guaranteed at home [77,78], access to teleconsultations provided from duly equipped local health centres [79] or the connectivity checked before conducting video consultations [27]. 

Multi-stakeholder efforts are required to develop interoperable platforms capable of providing equity in access to care, easily, without detriment to safety, in addition to providing a person-centred approach to care [73].

#### 3.3.5. Organisational Aspects

##### Quality of Neurology Teleconsultations

Some studies propose a generic approach to quality in the complementary use of TN interventions [65], from the monitoring of general quality criteria in the provision of health services [80], to the provision of TN in particular, from which two fundamental concepts are proposed: *service quality*, related to the ability of internet services to meet the technical requirements in terms of the connectivity indicated for users, and *experience quality*, related to the general acceptability of a service or application and linked to the extent to which the service in question meets the needs of users. In the latter, aspects of service quality, such as response time, reliability, security and service costs, are taken into account [44].

In the pursuit of quality care in the specific field of TN, the American Academy of Neurology (AAN) has developed a specific guideline to guide medical students, residents and practitioners in the safe practice of TN [81]. In addition, the AAN has formulated a series of practical recommendations in the *Telehealth Position Statement* [82] that include improving access and insurance coverage, ensuring fair reimbursement, reducing regulatory and legislative barriers, and expanding the evidence for TM by promoting research on its proper role and value in neurological care and on the costs associated with the provision of these TN services.

Another example is the *Movement Disorder Society Task Force on Technology* [70], which comprises members from clinical, academic and scientific institutions from several countries, those who have proposed a roadmap for the implementation of patient-centred digital outcome measures using mobile health technologies for Parkinson’s disease care. This collaborative effort will foster the development of integrated systems that can achieve more sophisticated characterisation of patient function, better tailoring of symptomatic therapy, increased patient engagement and self-assessment, and improved overall healthcare outcomes, as digital outcomes correlate with patient-centred global scales for appropriate domains [70].

In an analysis of 2589 telehealth consultations in neurologically affected children, performed during the COVID-19 pandemic, Rametta et al. [65] rated the vast majority of TN consultations as successful, with only a small proportion of consultations requiring short-term in-person follow-up. These results suggest that TN is feasible and effective for a large proportion of paediatric neurologic care.

#### 3.3.6. Environmental Aspects

A single study has been identified that evaluates environmental aspects related to TN service provision [52] evaluating the greenhouse gas (GHG) emissions related to air travel that is avoided when patients receive neuro-emergency telemedicine care. Whetten [52] compared the emissions related to the transfer of the patient in air ambulances from a rural hospital to a referral site (the rural hospital’s designated level-one trauma centre), compared to the GHG emissions associated with TN consultations. For the total visits, the calculation of the total travel distance avoided resulted in 618.77 metric tons of CO2 emissions avoided or 0.306 metric tons of CO2 per patient. The GHG emissions associated with the use of the TN equipment measured in electricity demand were calculated at 176 W for the use of a monitor and a standard personal computer, with a total time of 1241 h, which would generate 32 kg of CO2 emissions (considering the TM equipment and the patient’s equipment) for the total number of consultations evaluated.

The study concludes that the use of TN results in a minor effect compared to GHG emissions produced by patient transfers, and suggests that the methodology used in this study can be applied to other areas of TN that are intended to be implemented taking into account a carbon footprint reduction policy. However, the design of this study does not allow for a comparison of the health results obtained through the two healthcare alternatives.

### 3.4. Implementation Issues

Regarding the implementation of TN services, two key aspects emerged from the studies: *consultation and information recording times*, and *interdisciplinary coordination*.

At the *organizational level,* some studies point out, as one of the key aspects within the organization, the availability of a professional technical team capable of solving logistical, technical, legal and financial problems derived from the complementary use of TN consultations, as well as coordinating the training activities received by all clinical and administrative staff [53,55]. These support teams could provide communication between clinical team members reinforced by weekly update meetings on legal aspects, software licensing and best practices [53].

Workflow adjustments and technical support to integrate the platform with the electronic health record and to formalize financial reimbursements must also be assumed. Cloud storage has enabled sufficient expansion among many providers and thousands of simultaneous TN queries [53]. Other studies emphasize the importance of the availability and proper maintenance of device provisioning and connectivity [83].

Besides professional commitment [55], the training of the whole team, including videos, clinical practice guidelines and live sessions by specialists, has been highlighted [34]. It is considered that the use of TN should be part of the academic curriculum and training followed by resident physicians. As discussed above, the AAN Telemedicine Working Group formed a sub-working group to develop a framework for a formal TN curriculum in residency programmes, which it is hoped can also be introduced to medical students [81].

At the *patient and caregiver level*, in the video consultation cases during the COVID-19 pandemic in the elderly population for Parkinson’s disease and movement disorders, there were some problems with teleservices, related to the lack of digital and health literacy and the regulation of the participation of other family members in the TN consultation [54].

To ensure the successful implementation of complementary TN consultations, a preparatory process with patients beforehand has been recommended [54]. Instructions should be provided on basic aspects of the process, such as downloading the necessary applications and software for telematic consultation, as well as the location and use of alternative devices required for thorough clinical assessment [53]. Some patients may require support from caregivers or family members to access video consultations and telephone support for connection [64]. However, when technology places a significant burden on patients, interventions may have poor adherence—for example, if they require a regular input of symptoms or medication data [84].

## 4. Discussion

The objective of this review was to assess the ethical, legal, organisational, social and environmental aspects involved in the complementary use of TN as an adjunct to face-to-face neurology consultations. These aspects refer to an evaluation that goes beyond the core dimensions of conventional HTA (effectiveness and safety), taking into account individual, ethical and cultural aspects. The analysis of these dimensions is particularly relevant, due to two different arguments. The first one is the need to promote person-centred healthcare strategies focused on people’s needs and preferences. The second has been conditioned by the recent difficulties of the health systems, caused by the COVID-19 pandemic, in meeting the population’s health needs. In both cases, the use of technologies to promote remote health care reveals different organisational, legal, and ethical dilemmas that must be considered [4].

To incorporate research questions into this study that were relevant to all stakeholders, we introduced the VALIDATE [20,21,22] framework early in the review process. In the health research field, the criteria of patients or stakeholders, in general, might differ from those of researchers and healthcare professionals and can add value. Patients may have important insights that clinicians and researchers may miss, information about things that can cause problems for them or the types of technology and outcomes that patients value or are concerned about [85]. The participation of patients in the reorganisation of care processes and research can increase the perceived relevance and acceptability of the findings, which can lead to the research results being more fully applied. It has long been pointed out that there are several reasons for conducting participatory research [86]: acceptance is likely to improve, it generates important information, patients’ participation helps identify a whole list of perspectives and issues related to technology and participatory research leads to a better understanding of what technology does, while otherwise evaluation could be limited to measuring what a researcher, using a particular perspective, expects a technology to do.

Despite the diversity of methodological TN approaches reported, the different neurological conditions addressed, the type of technology used at the phase of care at which the service took place and the outcomes reported, the studies included in this review point to a few relevant themes. Firstly, issues related to equity of access, since it is a basic prerequisite of the TM. The digital divide and the limited health and digital literacy at the community level are both important barriers. TN, like TM in general, can pose challenges to accessing health care if it is offered as a non-voluntary proposal and without an alternative. In addition, older and disabled people may find it difficult to establish this relationship at a distance due to their lesser day-to-day relationship and cultural proximity to digital formats [59]. All these issues have been relevant to the consulted stakeholders, and the literature reviewed supports the need to guarantee these prerequisites to implement TN. The effectiveness of the organisational models at different points of contact along the care trajectory of neurological patients was another relevant concern for the panel members consulted. The results of this review do not, however, provide clear evidence about the effectiveness of these organizational models. In countries such as Belgium and France, TM reimbursement is regulated [76]; however, we have not found clearly defined specific legislation regulating the implementation of TN in the Spanish context, so there is some legal uncertainty [87,88,89]. The regulation of virtual medical activity must move forward to overcome the legal gap in which teleneurology care currently finds itself. While TN services can overcome geographical barriers and travel costs, the absence of specific legislation on TM and TN has been found to leave unresolved the problem of universal access to this service [90], as well as the inherent challenges of data protection and professional responsibilities [30,34,46,58,67,72], all of which creates a degree of uncertainty for healthcare professionals [89,91]. However, despite this absence of a specific legal framework for TN, important steps have been taken in the implementation of TN.

According to the Implementation Guide for Telemedicine Services published by the World Health Organisation (WHO) in 2022 [92], TM creates some special situations with unique considerations necessary to protect patient privacy and safety. All these considerations are perfectly applicable to the field of TN, as a particular type of TM. Legal regulations are currently a challenge and a key factor for the successful implementation and development of TM. These legal instruments provide a framework of security for the healthcare professional during the exercise of their clinical activities.

As TM system functionalities are established and refined, the WHO recommends the implementation of regulatory and protective arrangements for healthcare workers, patients and their health information, to mitigate risks and maximise end-user confidence in the system. In general, these regulatory aspects include the protection, privacy and accessibility of personal data, as well as the credibility of healthcare professionals providing TM services.

Special attention has been given to aspects related to the protection of the therapeutic relationship, both for the professional and for the patient, so that it can be developed respecting the ethical principles that govern its treatment [20,21,22,30,34,46,58,72]. The questions proposed by the working group also pointed out the concern for the legal aspects related to the implementation of TN in Spain. The review as carried out has identified this problem beyond the Spanish context, since a concern for the lack of regulation in different scenarios has been highlighted. Patients are increasingly willing to adopt TN systems to improve home care and daily self-management. An essential step towards wider adoption of these systems is to increase regulatory compliance, better define stakeholder responsibilities [88,93] and improve the necessary complementarity of TN with routine face-to-face care [94,95]. Evidence from studies shows that access to TN services can be limited by aspects related to a healthcare organisation, such as the poor reorganisation of roles and workflows and a lack of training for healthcare and administrative staff and patients. During the VALIDate process, it was noted as a concern that health professionals may need specialised training, ongoing support, and practical experience in performing teleneurology and that it may require a model of care delivered by health professionals with experience in this field.

The studies included in this review show that the provision of TM services generates a series of contradictions and tensions [4]. While it is true that there is evidence supporting the acceptability of TN for a proportion of patients and professionals and its feasibility at the time of implementation, other studies identify challenges or dilemmas that condition its acceptability and feasibility. Acceptability may be threatened by various aspects such as (1) the configuration and coverage of internet services, which result in connection failures, frequent interruptions, poor sound or image quality; (2) a lack of understanding of the functioning and benefits of the use of monitoring devices; (3) a perception of the loss of continuity of care and a relational distance and trust with the healthcare professional [27]. Specific training programmes for patients and professionals and the design of a comprehensive plan that fosters the commitment of both parties and establishes clear guidelines for the functioning of TN at home could contribute to the solution of these challenges that compromise the acceptability of the service [27].

In relation to healthcare professionals, the analysis has identified the risk of dehumanisation in the clinical relationship as an element that may influence the acceptability and even the feasibility of TN [71,72], as some consultations require direct contact for an adequate examination—hence the convenience of TN as a complementary service to regular face-to-face care is emphasised. This finding is consistent with some of the concerns expressed by the experts in the virtual meeting and was therefore included as a question to be addressed in this scoping review. TM is a welcome but disruptive innovation because it has the potential cost of altering the dynamics of physician–patient interaction, i.e., the tactile and other non-verbal cues that occur during human interaction in face-to-face settings. How physicians express empathy during TM encounters is an important but unexplored question, despite the abundant literature establishing the clinical efficacy of TN in acute stroke care, and its importance in maintaining patient trust [62]. Stakeholders also referred to socioeconomic and cultural barriers to access.

The findings show that at the practitioner level, the aspects that may affect acceptability are the technical quality of the service and organisational support. This has to do with having an interdisciplinary team to coordinate logistical, technical, legal and financial aspects derived from the use of TN, the training of clinical and administrative staff, and adequately equipping and maintaining the organisation’s devices and connectivity. Another key element reported by the studies analysed is the adaptation of existing clinical protocols [72,83], which allows professionals to develop technological and communication skills, as well as to fulfil basic practical aspects to be followed in TN consultations [53]. Similarly, the results show that to guarantee the successful implementation of the TN service, it is essential to follow a protocolised process of preparation with patients and companions/caregivers to guide patients before, during and after the virtual clinical encounter [48,53,55,57]. 

Concerning the implementation of TN, we found that the formats in which TN is carried out (synchronous or asynchronous, with professional or caregiver accompaniment) and the type of technology used must be related to the clinical objectives set, requiring users (both clinicians and patients) to have skills linked to ICTs and the organisation of the digital environment in which health interactions take place.

During the COVID-19 pandemic, numerous measures have been taken at the organisational level to ensure the quality of care and patient safety. Among the measures taken to reduce the risk of infection, visits to health centres have been reduced and virtual consultations have been increased [53,55,64,87]. However, the COVID-19 pandemic is unmasking an emerging form of technology-related social inequality: policy and community interventions are needed to support the most socially vulnerable populations and prevent social inequalities in health. In this sense, the panel of experts consulted stated that there could be a gap between the potential of TN in terms of the promise of access and efficiency and the reality of its implementation, during the COVID-19 pandemic and in the current period of cuts in health spending, in which the cost-effectiveness of the savings has not been taken into account.

Once the pandemic is over, the question will be whether TN should be limited to periods of a health crisis or whether it can become a new way of practising medicine. TN lacks specific standards and presents loopholes that leave physicians with a considerable degree of uncertainty [87].

Finally, the study of Whetten [52] represents a methodological contribution to the estimation of greenhouse gas (GHG) emissions that can be avoided due to TN services, taking into account that the health care sector is one of the sectors that contribute more to global warming. However, it should be considered as a limitation of the study that it does not present an evaluation of the GHG emissions generated by the implementation of the TN from a whole life-cycle approach to technology. Although the investigation is still limited, this study can serve as a reference to promote future research that seeks to evaluate the carbon footprint of the complementary service through TN, thus contributing to the mitigation of climate change and its consequences.

Planetary health and sustainability have been considered fundamental principles within the ethics of care to provide ethical and quality medical care within the framework of the Planning and Evaluation of Remote Consultation Services (PERCS) [4]. Despite the relevance and increasing attention given to the environmental impact of health technologies, this aspect is nonetheless complementary, concerning the main outcome measures of comparative effectiveness, safety and cost-effectiveness.

It should be noted that potential obstacles that contribute to limiting the development of a TM service, such as lack of legal clarity and the specific fragmentation of a given legal framework, can only be addressed through a coordinated approach between the different organisations that manage health resources [96].

This scoping review is framed in an HTA report, which has limited the deepening of some of the topics proposed by the stakeholders. Aspects that could not be addressed are identified as research needs, which will be the subject of future work, as reflected in Table 1. 

### Strengths and Limitations

The main strength of this review has been the contribution made by stakeholders in the initial phase of defining the research question in a virtual consensus meeting, developed as part of the process proposed by the VALIDATE project analysis framework [20,21,22]. This has allowed our results to gain relevance, as they reflect the different perspectives and values of the group of stakeholders consulted. This strength is, however, context-dependent to the Spanish setting because all the participants in the consultation were selected in this country.

Another potential strength of this article lies in the combined application of our theoretical framework. Indeed, the combined assessment of ethical, legal, organisational and social aspects through the EUnetHTA Core Model 3.0, the criteria established by the Spanish Network of Health Technology Assessment Agencies of the National Health System and the analysis of environmental dimension and the criteria of the European VALIDATE project offers new perspectives for the analysis of the context in scoping reviews and other types of reviews.

Regarding limitations, although a greater balance between the neurological disorders in the included studies would have been desirable, the available evidence reflects a greater interest in research on acute phases of some specific diseases, such as stroke, with a greater number of studies included in this review. 

Future work that could also be considered include an interpretive phenomenological approach to provide a deeper understanding of the experiences of patients and professionals, as there is considerable variation in individual perceptions of TN and how the program was integrated (or not). Participatory research in different conditions and clinical settings would contribute to a better understanding of the implications of TN implementation.

According to the report published by WHO in 2020 on the implementation of telemedicine services at the time of COVID-19, the following guiding principles should be considered for the successful implementation of telemedicine services [97]: patient-centred care, a multi-sectoral and multi-disciplinary approach, strong digital governance, equity and inclusion, usability and communication and contextualisation and localisation.

In addition, Kho et al. [98], suggest the need for a process-based approach that comprehensively deploys a combination of strategic and operational practices when managing change efforts. Rather than focusing on barriers and facilitators of change, they encourage future telemedicine research to examine the change processes and practices used to achieve successful implementation, particularly those practices that address cultural and people issues, as many barriers to adoption focus on attitudes and behaviours towards change.

## 5. Conclusions

Overall, the reported results reaffirm the necessary complementarity of TN with regular face-to-face care. This need for complementarity has to do with factors such as acceptability, feasibility, risk of dehumanisation and aspects related to the privacy and confidentiality of sensitive data. Likewise, the findings on implementation show that it is crucial to have an interdisciplinary team capable of coordinating logistical, technical, legal and financial aspects derived from the use of TN, as well as the training of clinical and administrative staff, professionals and patients. Balancing the interests and needs of professionals, and guaranteeing the connectivity and quality of all technological resources is also relevant.

TN should be performed in a framework of person-centred care, where co-design and humanised treatment can be guaranteed. It is a therapeutic relationship mediated by technology, where the same ethical, deontological and legal principles that guide face-to-face care must be followed even more carefully.

## Figures and Tables

**Figure 1 ijerph-20-03694-f001:**
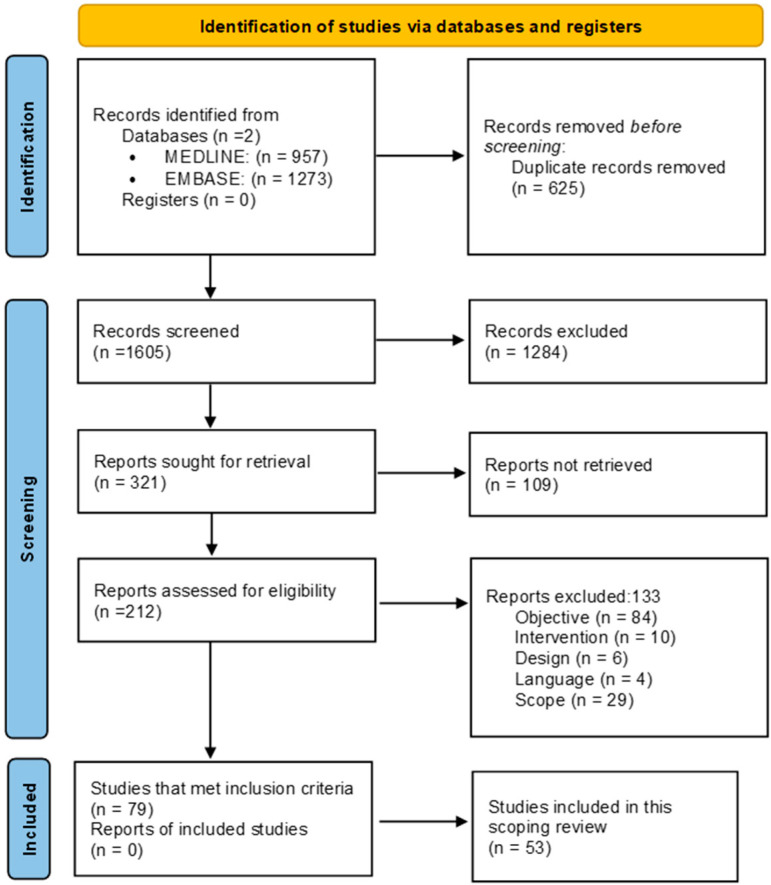
PRISMA flowchart of the study selection process.

**Table 1 ijerph-20-03694-t001:** Themes emerging from the virtual stakeholder meeting.

Questions Addressed within This Review	Research Needs
How can/should universal accessibility of teleneurology be ensured? Voluntariness at the European level, testing of connectivity and compatibility of devices. What access barriers exist for access to teleneurology? How can these access barriers be reduced? What technologies and ways of implementing these technologies or interventions facilitate access?	Could teleconsultation be positive for patients with behavioural or cognitive problems? Does it reduce the time between visits for patients? Is it more efficient for professionals?
	How effective are the different organisational models for contact points?
How should patient perspectives and preferences be considered? Do some patients feel insecure about teleneurology or perceive it as an inferior model of care to traditional models, such as face-to-face care?	
	How can the quality of teleconsultation be ensured? Should quality criteria be changed or adapted to teleneurology?
- How can/should a teleservice be organized taking into account consultation and recording times adapted to each type of contact? - Can increased multidisciplinary coordination be managed efficiently and cost-effectively?	Can the time management of healthcare professionals be improved by promoting work-life balance?
How should privacy and confidentiality be ensured in the different types of teleconsultation and in the storage of additional data?	
- How can it be determined that teleneurology is used appropriately as a complement to face-to-face consultations? Which types of healthcare events require face-to-face attendance and which ones do not? - Could a checklist in the report help in deciding when it is appropriate to implement/pilot complementary teleneurology?	
How can continuity of care be ensured in the face of possible connection or device failures by healthcare providers and patients?	

## Data Availability

The data presented in this study are available in Appendix A.

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
