# Peer review of "Ethical, Legal, Organisational and Social Issues of Teleneurology: A Scoping Review"

_ijerph, 2023, doi:10.3390/ijerph20043694_

Round 1
Reviewer 1 Report
General comments
The topic of the manuscript is interesting and relevant, as it is important to continuously monitor and control the quality of applications of teleneurology, as well as other telemedicine implementations. After some revisions, this manuscript may provide an important contribution to this subject area.
Specific comments and suggestions for improvement
1) The following five points are about the consistency of research aim/research question:
a) Section 1, p. 2, lines 74 - 86: Strictly speaking, the acronym ELSI includes ethical, legal, and social issues (or acronym ELSA = ethical, legal, and social aspects).
In your review, you have added organisational and environmental aspects, and changed social issues to patient perspective. This is not a problem per se, but it would be beneficial if you can explain/justify this modification, specifically in the context of teleneurology.
b) In section 2, page 2, lines 91 and 92, you refer to assessment of ethical, legal, organisational, and social aspects.
c) In section 3, p. 5, lines 170-172, the research question includes “ethical, legal, organisational and social aspects”.
d) In #2 and #3 above, is the term “social aspects” synonymous with “patient perspective”? Social aspects are often interpreted as social relations between individuals and shared features within a community, whereas patient perspective refer to an individual perception.
e) In section 4 Discussion, the terms are ethical, legal, organisational and social aspects.
2) Table 1: section 2.1, p. 3, line 120: It is not quite clear how the different columns are related to each other:
· Are the questions for the working group (column #1) informing the questions addressed in the review (column #2)?
· How are the questions in the last column, research needs, derived from the two others, and are these questions also included in the review?
· Are the questions for the working group (column #1) also the themes emerging from the meeting?
3) The following two points are about the number of included studies
a) Section 3: p. 5, Figure 1 and lines 167-169: What is the number of the studies included in the review? In the figure, the number is n=79, but in the text, it is stated that the only 53 studies are included in the review. The latter number seems to be consistent with the Table 2 (table in the supplementary material).
Suggestion to avoid confusion: In figure 1, you could refer to n=79 as studies that met the inclusion criteria, and maybe also add a box, where the final number of included studies (n=53) is shown.
b) Please add a short explanation/justification for why the 26 studies (related to feasibility) were excluded from the review, even if they met the inclusion criteria.
4) Is Table 2 the table that can be found as a supplementary Word-file? Please clarify this and add a table caption.
5) In section 3.3, line 16 (the numbering seems to re-start after table 2) you define the second issue in the main findings as “Utility”. However, in section 3.3.2 (p. 2, line 50) this issue is named “Usability”. They are both important aspects of UX, but according to Nielsen there is a slight difference, as usability is about ease of use, whereas utility is about whether the functions are needed. The combination of the two concepts determines the system’s usefulness.
In section 3.3.2 you seem mainly to describe usability issues, so I’d suggest that you either change the term in section 3.3 to Usability, or use Usefulness as an umbrella term, in both sections 3.3 and 3.3.2.
6) Section 3.3.4, p. 3, line 118 (the numbering seems to re-start after table 2): “People with different moderate and severe physical and mental abilities who, …” should probably be “physical and mental disabilities”
7) The numbering of the sections and there heading levels are not quite consistent:
· Section 3.4 seems to be missing
· Section 3.5 Organisational aspects is part of the list of main findings, so it should probably be numbered 3.3.5, and the following sections 3.3.5.1 Quality of neurology teleconsultations, 3.3.5.2 Implementation issues.
· If I understand correctly, you consider environmental aspects as part of the Organisational aspects. However, environmental issues, such as CO2 emission, have broader societal impact, also outside the health care organisation. Please consider if environmental aspects should be a separate, independent category. In this case, this section should be numbered 3.3.6.
8) Informed consent statement: You don’t need this for the scoping review, but how did you obtain informed consent from the participants in the virtual stakeholder meeting?
9) Minor translation errors:
Table 1, p. 3: The first question in the column “Research needs” has two question marks.
line 432: Reference list should have title “References”
Some of the references contain dates that should be translated to English, for instance
line 435: 1 de marzo de 2020; line 439, 440: octubre de 2021; etc.
Author Response
Specific comments and suggestions for improvement
Thank you for your review and comments. We have improved our work.
1) The following five points are about the consistency of research aim/research question:
- a) Section 1, p. 2, lines 74 - 86: Strictly speaking, the acronym ELSI includes ethical, legal, and social issues (or acronym ELSA = ethical, legal, and social aspects).
In your review, you have added organisational and environmental aspects, and changed social issues to patient perspective. This is not a problem per se, but it would be beneficial if you can explain/justify this modification, specifically in the context of teleneurology.
I agree with your comment. We cited the ELSI dimensions as a possible perspective, but we consider that the analysis should be broader and therefore we adopt for this review the Core Model 3.0 of EUnetHTA and the criteria established by Spanish Network of Agencies for Health Technology Assessment for the National Health Service. This theoretical framework includes the organizational perspective. In addition, we have also added the environmental dimension. We have added a couple of lines explaining this modification. Lines: 85-88.
- b) In section 2, page 2, lines 91 and 92, you refer to assessment of ethical, legal, organisational, and social
This paragraph is now better understood with the clarification made previously. We will be referring hereinafter to ethical, legal, organizational, social and environmental aspects
- c) In section 3, p. 5, lines 170-172, the research question includes “ethical, legal, organisational and socialaspects”.
This paragraph is now better understood with the clarification made previously.
- d) In #2 and #3 above, is the term “social aspects” synonymous with “patient perspective”? Social aspects are often interpreted as social relations between individuals and shared features within a community, whereas patient perspective refer to an individual perception.
As you point out, these are different dimensions. In this review we have chosen to focus as a more general category on the social aspects, leaving the patient perspective as a subcategory within the social, where appropriate.
- e) In section 4 Discussion, the terms are ethical, legal, organisational and social
This is correct, taking into account the clarifications made previously. The reference to ELSI has been deleted.
2) Table 1: section 2.1, p. 3, line 120: It is not quite clear how the different columns are related to each other:
- Are the questions for the working group (column #1) informing the questions addressed in the review (column #2)?.
Column 1 has been removed because it can be confusing. In the text we explain what column 2 refers to. It should be noted that this table now has only two columns.
- How are the questions in the last column, research needs, derived from the two others, and are these questions also included in the review?
An explanation has been added so that it is understood where the column on Research needs comes from. Lines 122-24. These topics have not been included in this revision.
- Are the questions for the working group (column #1) also the themes emerging from the meeting?.
They are not. In any case, column 1 has been eliminated.
3) The following two points are about the number of included studies
- a) Section 3: p. 5, Figure 1 and lines 167-169: What is the number of the studies included in the review? In the figure, the number is n=79, but in the text, it is stated that the only 53 studies are included in the review. The latter number seems to be consistent with the Table 2 (table in the supplementary material).
Suggestion to avoid confusion: In figure 1, you could refer to n=79 as studies that met the inclusion criteria, and maybe also add a box, where the final number of included studies (n=53) is shown.
We have added in the flow chart an explanation that 79 studies are those that met the criteria and 53 studies are the studies included in this review
- b) Please add a short explanation/justification for why the 26 studies (related to feasibility) were excluded from the review, even if they met the inclusion criteria.
Although those 26 studies also met the criteria, they were not included in this review because they did not fully fit the scope of this review. Furthermore, given that those studies that are more related to implementation, they will be included in a review dedicated to this topic.
4) Is Table 2 the table that can be found as a supplementary Word-file? Please clarify this and add a table caption.
The original idea was that Table 2 would be included directly in the text. Since it is a large file of 6 pages that can make the manuscript difficult to read, we have decided to include it as additional file 4.
5) In section 3.3, line 16 (the numbering seems to re-start after table 2) you define the second issue in the main findings as “Utility”. However, in section 3.3.2 (p. 2, line 50) this issue is named “Usability”. They are both important aspects of UX, but according to Nielsen there is a slight difference, as usability is about ease of use, whereas utility is about whether the functions are needed. The combination of the two concepts determines the system’s usefulness.
In section 3.3.2 you seem mainly to describe usability issues, so I’d suggest that you either change the term in section 3.3 to Usability, or use Usefulness as an umbrella term, in both sections 3.3 and 3.3.2.
It was indeed a mistake, when we say "utility" in line 16, we meant "usability". This has been corrected
6) Section 3.3.4, p. 3, line 118 (the numbering seems to re-start after table 2): “People with different moderate and severe physical and mental abilities who, …” should probably be “physical and mental disabilities”.
Thank you, it was indeed a mistake, we refer to mental disabilities.
7) The numbering of the sections and there heading levels are not quite consistent:
- Section 3.4 seems to be missing.
Thank you, section 3.4 was indeed missing. We have corrected the numbering of the headings
- Section 3.5 Organisational aspectsis part of the list of main findings, so it should probably be numbered 3.3.5, and the following sections 3.3.5.1 Quality of neurology teleconsultations, 3.3.5.2 Implementation issues.
According to your suggestion, we have changed it.
- If I understand correctly, you consider environmental aspects as part of the Organisational aspects. However, environmental issues, such as CO2emission, have broader societal impact, also outside the health care organisation. Please consider if environmental aspects should be a separate, independent category. In this case, this section should be numbered 3.3.6.
According to your suggestion, we have changed it.
8) Informed consent statement: You don’t need this for the scoping review, but how did you obtain informed consent from the participants in the virtual stakeholder meeting?
These people signed a consent form to participate in the health technology assessment report, so it was not necessary to have a specific one for this online meeting. We add an explanation about it. Lines 121-126
9) Minor translation errors:
Table 1, p. 3: The first question in the column “Research needs” has two question marks.
We have corrected it.
line 432: Reference list should have title “References”.
We have corrected it.
Some of the references contain dates that should be translated to English, for instance
line 435: 1 de marzo de 2020; line 439, 440: octubre de 2021; etc.
We have corrected it.

Reviewer 2 Report
Title: Ethical, legal, organisational and social issues of Teleneurology. A scoping review
· A brief comment regarding the title; it should not use a period (.). I propose that the title is as follows: “Ethical, legal, organisational and social issues of Teleneurology: A scoping review”.
· Abstract and introduction
o Please edit the following sentence as it lacks clarity: “More than 25 years ago, the increase in the population demand for care, together with territorial inequalities in patient access to health services, and the difficulties of economic sustainability of health systems in most countries, found in telemedicine an alternative to accommodate potential solutions to these three divergent challenges” perhaps remove ‘in’ in this sentence? (p. 2 line 44-47).
o And please check the following sentence: ‘Unfortunately, for almost two decades, validity limitations in available scientific evidence delayed the widespread adoption of the different modalities of telemedicine in its different applications. Time has elapsed, accompanied by improvements in the designs of evaluative studies on telemedicine, and, more recently, the need of tools to meet the population health needs during the Covid-19 pandemic, has favoured the implementation and diffusion of telemedicine in all health fields [3]’ (p. 2 line 48-53). I suggest removing ‘validity’ and change ‘the need of’ to ‘the need for’.
o Please do a thorough editing for English language including usage of articles such as ‘the’ in the manuscript. And ensure that it is either in American or Australian English. Consistency is key.
o And add a ‘map’ of the article in the last paragraph in the introduction; this kind of ‘map’ would explain section 2. will be about methods and so on.
o Please specify the objectives of your scoping review, as it stands, there are no explicit objectives. The following article could be referred to as a reference: https://bmchealthservres.biomedcentral.com/articles/10.1186/1472-6963-14-271.
· Methods
o I suggest removing ‘materials’ in the ‘materials and methods’ (P. 2 L.87-88)
o Please explain how participants for an online meeting on 6 June 2021 were selected.
o Please specify the kinds of neurological disease covered in this scoping review.
o Was there a protocol used? And if there were, was it registered in Prospero? The following article might help in explaining protocol, in case needed: https://bmcmedresmethodol.biomedcentral.com/articles/10.1186/1471-2288-12-114.
· Discussion section:
o Please tidy up p. 5 line 182-192 – as of now it is not organised well. I think there is a typo as well.
o I would appreciate it if there is an explanation as to why ‘the EUnetHTA Core Model 3.0 framework’ is used in your study, including justifications for this framework.
o Does this scoping review also looks at the extent to which TN (tele neurology) tools are inclusive for people with mental health disabilities in the existing studies? If not then please exclude this from your study.
o As of now, there is a lack of integration methodologically and in terms of findings from the online discussion and the scoping review activity in the discussion section – I suggest this be improved and be made more intertwined in the discussion section.
o Do any of these guidelines mentioned on page 4 talk about patient safety in TN used in neurological diseases?
o How does the EU level deal with risks that arise from the legal uncertainty in the regulation of TN? This question is as a response of statement on page 6 L267-270.
o What are other strengths of this article/manuscript? And based on your findings please exemplify future directions in terms of health services via TN for neurological diseases and rehabilitation globally and in Europe.
· Overall, I find findings of this scoping review (study) to be convincing. This is a very interesting study.
Author Response
A brief comment regarding the title; it should not use a period (.). I propose that the title is as follows: “Ethical, legal, organisational and social issues of Teleneurology: A scoping review”.
We have modified the title.
Abstract and introduction
Please edit the following sentence as it lacks clarity: “More than 25 years ago, the increase in the population demand for care, together with territorial inequalities in patient access to health services, and the difficulties of economic sustainability of health systems in most countries, found in telemedicine an alternative to accommodate potential solutions to these three divergent challenges” perhaps remove ‘in’ in this sentence? (p. 2 line 44-47).
Done. We have shortened the paragraph.
o And please check the following sentence: ‘Unfortunately, for almost two decades, validity limitations in available scientific evidence delayed the widespread adoption of the different modalities of telemedicine in its different applications. Time has elapsed, accompanied by improvements in the designs of evaluative studies on telemedicine, and, more recently, the need of tools to meet the population health needs during the Covid-19 pandemic, has favoured the implementation and diffusion of telemedicine in all health fields [3]’ (p. 2 line 48-53). I suggest removing ‘validity’ and change ‘the need of’ to ‘the need for’.
Done
o Please do a thorough editing for English language including usage of articles such as ‘the’ in the manuscript. And ensure that it is either in American or Australian English. Consistency is key.
Done
o And add a ‘map’ of the article in the last paragraph in the introduction; this kind of ‘map’ would explain section 2. will be about methods and so on.
Done. We have included this paragraph at the end of the introduction.
o Please specify the objectives of your scoping review, as it stands, there are no explicit objectives. The following article could be referred to as a reference: https://bmchealthservres.biomedcentral.com/articles/10.1186/1472-6963-14-271.
Done. The objective has been specified
- Methods
o I suggest removing ‘materials’ in the ‘materials and methods’ (P. 2 L.87-88).
Done
o Please explain how participants for an online meeting on 6 June 2021 were selected.
These experts were part of the working group and collaborators to the health technology assessment report, who were contacted for their experience and relevant publications in the field of teleneurology. We add an explanation about it. Lines 121-126.
o Please specify the kind of neurological disease covered in this scoping review.
The different types of neurological diseases treated in this study are described in the section "Characteristics of the studies".
Was there a protocol used? And if there were, was it registered in Prospero? The following article might help in explaining protocol, in case needed: https://bmcmedresmethodol.biomedcentral.com/articles/10.1186/1471-2288-12-114.
Only the protocol for the study related to the effectiveness and safety of the complementary use of teleneurology in routine practice has been registered, with the following registration number: CRD42021262578.
Discussion section:
Please tidy up p. 5 line 182-192 – as of now it is not organised well. I think there is a typo as well. Done.
This paragraph has been corrected.
I would appreciate it if there is an explanation as to why ‘the EUnetHTA Core Model 3.0 framework’ is used in your study, including justifications for this framework.
We use the EunetHTA 3.0 Core Model because in the field of HTA, approaches to analyse the impact of the use of health technologies on the ethical, legal, organisational and social dimensions are recommended.
Does this scoping review also looks at the extent to which TN (tele neurology) tools are inclusive for people with mental health disabilities in the existing studies? If not then please exclude this from your study.
It does not refer to mental disability, we have corrected it by replacing mental by cognitive and added the example of multiple sclerosis MS. Lines 345-347.
o As of now, there is a lack of integration methodologically and in terms of findings from the online discussion and the scoping review activity in the discussion section – I suggest this be improved and be made more intertwined in the discussion section.
We have improved the discussion, taking up specific aspects that arose in the virtual meeting with the experts. Everything has been marked with change control in the discussion section.
How does the EU level deal with risks that arise from the legal uncertainty in the regulation of TN? This question is as a response of statement on page 6 L267-270.
According to the Implementation Guide for Telemedicine (TM)Services published by the World Health Organisation (WHO) in 2022, TM creates some special situations with unique considerations necessary to protect patient privacy and safety. All these considerations are perfectly applicable to the field of TN.
Legal regulations during the implementation and development of telemedicine services are a challenge and a key factor in the success of TM. These legal instruments provide a framework of security for the healthcare professional during the exercise of their clinical activities.
As TM system functionalities are established and refined, the WHO recommends the implementation of regulatory and protective arrangements for healthcare workers, patients and their health information, to mitigate risks and maximise end-user confidence in the system.
In general, these regulatory aspects include the protection, privacy and accessibility of personal data, as well as the credibility of healthcare professionals providing telemedicine services.
It should be noted that potential obstacles that contribute to limiting the development of a telemedicine service, such as lack of legal clarity and the specific fragmentation of a given legal framework, can only be addressed through a coordinated approach between the different organisations that manage health resources.
What are other strengths of this article/manuscript?
The following paragraph has been included in the strengths and weaknesses section.
Another potential strength of this article lies in the combined application of our theoretical framework. Indeed, the combined assessment of ethical, legal, organisational and social aspects through the EUnetHTA Core Model 3.0, the criteria established by the Spanish Network of Health Technology Assessment Agencies of the National Health System and the analysis criteria of the European Validate project, offers new perspectives to the analysis of the context in scoping reviews and other types of reviews.
- And based on your findings please exemplify future directions in terms of health services via TN for neurological diseases and rehabilitation globally and in Europe.
We have added a paragraph at the end of the discussion on implications for future practice.
According to the report published by WHO in 2020 on the implementation of telemedicine services at the time of COVID-19, the following guiding principles should be considered for the successful implementation of telemedicine services: patient-centred care, multi-sectoral and multi-disciplinary approach, strong digital governance, equity and inclusion, usability and communication and contextualisation and localisation.
In addition, Kho et al. suggest the need for a process-based approach that comprehensively deploys a combination of strategic and operational practices when managing change efforts. Rather than focusing on barriers and facilitators of change, they encourage future telemedicine research to examine the change processes and practices used to achieve successful implementation, particularly those practices that address cultural and people issues, as many barriers to adoption focus on attitudes and behaviours towards change.
Overall, I find findings of this scoping review (study) to be convincing. This is a very interesting study.
Thank you for your review and comments. We have improved our work.

Round 2
Reviewer 2 Report
- Apologies, but ‘Teleneurology’ should be written as ‘teleneurology’ (not with a capitalised ‘t’).
- Conclusion section is missing from the ‘map’ in the introduction section. Please add this.
- Thank you for adding an explanation on lines 121-126 p.3. ‘Stakeholders’ should be written as ‘stakeholders’.
- I still see inconsistent usage of Australian English throughout the text; please make sure it is consistent.
- The findings of this review have been shown to be insufficient to answer this question. P.12 Line 507-508: What question would that be? Please add more information for the sake of clarity.
- Thank you very much for making revisions for your manuscript.
-
Author Response
Response to reviewer (Round 2)
- Apologies, but ‘Teleneurology’ should be written as ‘teleneurology’ (not with a capitalised ‘t’).
It has been corrected
- Conclusion section is missing from the ‘map’ in the introduction section. Please add this
It has been added
- Thank you for adding an explanation on lines 121-126 p.3. ‘Stakeholders’ should be written as ‘stakeholders’.
It has been corrected
- I still see inconsistent usage of Australian English throughout the text; please make sure it is consistent.
English has been corrected throughout the text.
- The findings of this review have been shown to be insufficient to answer this question. P.12 Line 507-508: What question would that be? Please add more information for the sake of clarity.
Despite the absence of a specific legal framework for TN, progress has been made in the implementation of TN. However, it is still a challenge to make progress in terms of legal security and the digital divide. Some sections that indicate this have been highlighted in yellow in the discussion section.
- Thank you very much for making revisions for your manuscript.
Thank you for your comments and contributions, we have improved our paper.